# The Ecology and Phylogeny of Hosts Drive the Enzootic Infection Cycles of Hantaviruses

**DOI:** 10.3390/v11070671

**Published:** 2019-07-23

**Authors:** Matthew T. Milholland, Iván Castro-Arellano, Gabriel E. Garcia-Peña, James N. Mills

**Affiliations:** 1College of Agriculture and Natural Resources-Department of Environmental Sciences and Technology, University of Maryland, College Park, MD 1433, USA; 2United States Department of Agriculture-Agriculture Research Service, Invasive Insect Biocontrol and Behavior Laboratory, Beltsville, MD 20705, USA; 3Department of Biology, Texas State University, San Marcos, TX 78666, USA; 4Facultad de Medicina Veterinaria y Zootecnia, Universidad Nacional Autónoma de México, México City 04510, México; 5Centro de Ciencias de la Complejidad C3, Universidad Nacional Autónoma de México, México City 04510, México; 6UMR MIVEGEC, Maladies Infectieuses et Vecteurs: Ecologie, Génétique, Evolution et Contrôle, UMR 5290, CNRIS-IRD-Université de Montpellier, Centre de Recherche IRD, Montpellier Cedex 5 34192, France; 7Population Biology, Ecology, and Evolution Program, Emory University, Atlanta, GA 30322, USA

**Keywords:** hantavirus, zoonosis, virus ecology, rodents, shrews, phylogenetic modeling

## Abstract

Hantaviruses (Family: Hantaviridae; genus: *Orthohantavirus*) and their associated human diseases occur globally and differ according to their geographic distribution. The structure of small mammal assemblages and phylogenetic relatedness among host species are suggested as strong drivers for the maintenance and spread of hantavirus infections in small mammals. We developed predictive models for hantavirus infection prevalence in rodent assemblages using defined ecological correlates from our current knowledge of hantavirus-host distributions to provide predictive models at the global and continental scale. We utilized data from published research between 1971–2014 and determined the biological and ecological characteristics of small mammal assemblages to predict the prevalence of hantavirus infections. These models are useful in predicting hantavirus disease outbreaks based on environmental and biological information obtained through the surveillance of rodents.

## 1. Introduction

Hantaviruses (order: Bunyavirales; family: Hantaviridae; genus: *Orthohantavirus*) [1] are among the most widely distributed emerging pathogens known to date [2] and are found on every continent except Antarctica [3]. Currently, there are 38 genera with 60 orthohantaviruses (hereafter, “hantavirus”) described worldwide and officially recognized by the International Committee on Taxonomy of Viruses [1]. At least 22 of these viruses are known to be pathogenic to humans [4,5]. Small mammals including rodents (order Rodentia), serve as hantavirus hosts by maintaining and amplifying these pathogens in nature with no apparent signs of disease, thus becoming a source of infection that spreads these viruses to spillover hosts, including humans [6].

The geographic distributions of hantaviruses are dependent upon the natural history of their hosts [5]. Rodents of the families Cricetidae and Muridae are the principal known hosts of hantaviruses [6,7]. To date, hantaviruses hosted by multiple species in these two families are the only known serotypes causing disease in humans [8]. Hantavirus-antibody-positive individuals from species in the rodent families Heteromyidae, Nesomyidae, and Sciuridae have also been documented, thus suggesting these families might also serve as hosts [5,9,10]. Some hantaviruses have close associations with small mammal groups outside of the order Rodentia. These include the shrews (order Eulipotyphlya; family Soricidae) [11,12,13], moles (order Eulipotyphla; family Talpidae) [6,14], and bats (order Chiroptera; families Rhinolophidae, Nycteridae, and Vespertilionidae) [15,16].

Intraspecies transmission typically occurs horizontally via aggressive behavior, often associated with territory defense among adult males [17,18]. Rodents become chronically infected and show almost no clinical signs despite the presence of antibodies [19,20]. Vertical transmission is thought to be altogether absent or negligible in both wild and experimental environments [21,22], however, it has been documented in cotton rats (*Sigmodon hispidus*) [23,24]. Spillover hantaviral infections are accidental in secondary, dead-end hosts including humans and other primates [25], domestic and sylvan animals [26,27], lagomorphs [28], and marsupials [7]. However, assemblage-level ecological factors facilitating spillover events (e.g., species identity and relative abundance), particularly at the human–wildlife interface, lack empirically driven predictive models [29,30]. The prevention of these disease outbreaks requires the surveillance of antibody-prevalence in animal populations [31,32] and the development of accurate predictive models [33].

Other factors driving the maintenance and spread of hantavirus infections in rodent populations include the structure of small mammal assemblages [34,35]. The prevalence of a hantavirus infection in a small mammal assemblage generally decreases with increased species diversity of the assemblage. Thus, it has been hypothesized that increases in species richness decrease pathogen transmission [36]. The resulting “dilution effect” [37] is typically attributed to a relationship between the dominance of generalist host species and the reduced abundance of rarer species comprising the assemblage [19,38]. This phenomenon is thought to occur when increased assemblage diversity suppresses the density of the main host species and thus dilutes disease prevalence and density-dependent transmission events [37,39]. However, an opposite “amplification effect” is also known to occur when there is an increase in the transmission rate in assemblages with higher diversity [40]. Moreover, these two effects may occur concurrently in the same pathogen system, with the net effect of species diversity being determined by the strength of the competing mechanisms [40].

It may be possible to predict hantavirus-related disease risk in humans by utilizing assemblage diversity estimates of host species [41] and species identity indices such as phylogenetic predictors [31,34]. These relationships have been studied in some zoonotic systems [35,42], including specific hantavirus-related human infections [43,44,45] and particular host species [46,47]. However, the general form of a predictive model for hantavirus infection has not yet been thoroughly addressed [20].

Here, we developed predictive models for the prevalence of hantavirus infections in small mammal assemblages by using ecological assemblage correlates and accounting for the phylogenetic relationships between potential host species. To calibrate the models with prior information, we analyzed data derived from the scientific literature. We hypothesized that small mammal assemblages with more phylogenetically diverse species would have fewer hantavirus antibody positive individuals, where this relationship may be mediated by the differences in rodent host species abundance between low and high species-richness assemblages. Furthermore, we expected that assemblages with a higher phylogenetic lineage divergence would contribute to a decrease in hantavirus antibody prevalence, as interspecific transmission is less probable.

## 2. Materials and Methods

### 2.1. Site Data Collection

Species abundance distributions for rodent assemblages worldwide were gathered from peer-reviewed publications by using a combination of terms such as “Hantavirus, rodents, Hantavirus Cardiopulmonary Syndrome (HCPS), Hantavirus hemorrhagic fever with renal syndrome (HFRS), nephropathia epidemica (NE), hosts, and reservoirs” for articles published from 1971 to 2014 using Google, Google Scholar, Web of Science, and the US Centers for Disease Control and Prevention (CDC) homepage (www.cdc.gov). The CDC website provided relevant articles and related links including PubMed (www.pubmed.gov) and the U.S. National Library of Medicine (www.nlm.nih.gov). References listed in journal articles were also utilized to source the initial reports of hantavirus hosts, their related serotypes, and any human disease associations. Site data were categorically grouped by continent (i.e., Asia, Europe, North America, and South America), as reports of hantavirus surveys are often followed by human disease outbreaks and are monitored differently according to geographic and political regions [20]. For inclusion in the study, all individuals within each assemblage must have been trapped in Sherman live traps or snap traps and tested for hantavirus antibodies, revealing at least one positive individual reported at each site. Additionally, for model analyses, we included only sites reporting a raw abundance equal to or greater than 20 individuals and no greater than 200 individuals in the sampling effort (20 ≤ *n* ≤ 200; statistical rationale described below).

The following data were collected for each site: 1) the geographic identity (e.g., political location, latitude and longitude); 2) year and month of trapping or collection; and 3) abundance and identity of each species trapped or collected, including the number and identity of seropositive individuals. The specific geographic identity (i.e., site) was constrained, and sites were limited to trapping transects within an area no greater than 5 km² and a trapping effort which occurred within a 12-month period. Additionally, a type-specific hantavirus infection was recorded if the genetic identity of the hantavirus was confirmed by reverse transcription polymerase chain reaction (RT-PCR). Hantavirus identity information was used for reference only.

### 2.2. Response and Predictor Variables

Assemblage antibody prevalence was calculated as the average of within species prevalences across all species comprising an assemblage [48,49]. Using this calculation for prevalence provided a more accurate representation of the contribution of each species to the overall assemblage prevalence. This response variable was labeled the total assemblage seroprevalence (TAS) and was calculated for each site. Additionally, the antibody prevalence of the most abundant primary host [6] within each assemblage served as a second tested response variable (most dominant host seroprevalence, MDHS). TAS and MDHS predictive models were then compared using the statistical criterion described below.

Eleven diversity (e.g., assemblage) descriptors were gathered or calculated for all sites. These assemblage characteristics served as predictor variables against the response variables for seroprevalence: TAS and MDHS. These predictor variables included: 1) raw total abundance *(n),* or the number of individuals for all species reported for the assemblage; 2) raw species richness *(s),* the total number of species recorded at each site; and 3) estimated species richness (s_est_), which was also used because sampling efforts can limit or bias the reported number of species. To compensate for this potential bias, the Chao estimate (CHAO1) and the abundance coverage-based estimate of species richness (ACE) were calculated using the EstimateS programs [50,51]. These estimated values were compared to the raw species richness with a Student’s t-test to determine if any difference existed for use in the analysis. Variable 4) was the Shannon diversity index (H’), which was calculated using EcoSim 700 [52]. A benefit of using H’ is that, collectively among sites, it generally follows a normal distribution [53]. Variable 5) was the Hurlbert’s Probability of Interspecific Encounter (PIE) index, which is the probability that two randomly sampled individuals from an assemblage pertain to different species [54]. This evenness index, or measure of heterogeneity [55], was used to combine species richness and dominance characteristics of the assemblage [56]. We used EcoSim 700 to calculate PIE [52]. Variable 6) consisted of the mean nearest taxon distance and mean pairwise distance, which were phylogenetic hypotheses created in R (package “Picante”, http://artax.karlin.mff.cuni.cz/r-help/library/spacodiR/html/00Index.html/) using a cytochrome-b (Cyt-b) phylogeny created with data obtained from GenBank (http://www.ncbi.nlm.nih.gov/genbank/) comprising all species included in the generated dataset (“MrBayes”; http://nbisweden.github.io/MrBayes/). This phylogeny was used to generate predictor variables (mean nearest taxon distance, MNTD; mean pairwise distance, MPD) from the relatedness of each species within an assemblage using the “bladj” algorithm of phylocom (http://phylodiversity.net/phylocom/) [57,58,59]. The MNTD and variable 7)—the MPD values—were compared with a Student’s t-test to determine if any differences exist for use in the analysis. Variable 8) consisted of the phylogenetic diversity totals (PD), which relate to the minimum total length of all phylogenetic branches required to span a given set of taxa on the phylogenetic tree [60]. The bifurcating phylogeny branch lengths were calibrated to *x*-million years before the present divergence and PD totals estimated as Faith’s PD [61]. This phylogenetic species variability index was also based on cladistic information, and this index quantified the variability among species composing an assemblage [62]. PD was summarized using a matrix of the pairwise distances between taxa using Picante for R [61,63]. Variable 9) was an ordinate rank (rk), which was used where each species was given a discrete rank variable (from 1 to s) in decreasing values of the reported (n) within each assemblage. The rank variable for the assemblage was representative of the rank of the most dominant host within that assemblage. Rodents carrying hantaviruses known to cause disease in humans were listed as “reservoirs.” Variable 10) was the rarity threshold (R_t_), which was a classification of how common a species is from the sum of all species collected in each assemblage. With this categorical predictor variable, “rare” species were determined as having a relative abundance below the rarity threshold, and (R_t_) was calculated as (n/s). Species whose relative abundance was greater than this ratio were categorically listed as “abundant.” Variable 11) was the Berger-Parker dominance index (BP), which was determined by the inverse of the proportional abundance of the most abundant species: (d = 1/(n_max_/N)), where (n_max_) = the number of individuals in the most abundant species and (N) was the total number of individuals in the assemblage [53]. This index was used to describe the dominance component of the host or reservoir with the highest antibody prevalence. In the instance where multiple host/reservoir species were seropositive in an assemblage, the (*n*)-value and the site-specific attributes were used as context to determine the dominant species component.

### 2.3. Statistical Analysis

The relationship(s) between the response variables (seroprevalence calculated as TAS or MDHS) of the assemblage to the predictor variable(s) mentioned above were analyzed through stepwise linear regression models in R version 3.5.2 and the “leaps” package [64]. Because of the way the predictor variables are derived, correlations between predictor variables were analyzed using the Pearson product–moment correlation (Pearson’s r). Where correlations were present, principal component analysis (PCA) (e.g., dimensionality reduction) and the R package “FactoMineR” [65] were used to unify, or compress, correlated variables into a single index (i.e., eigenvectors), thus reducing the statistical issues of multicollinearity [66,67]. Additionally, all response and predictor variables were scrutinized, and violations in homoscedasticity were determined and corrected to fit a normal distribution. Models were compared using the Akaike Information Criterion (AIC_c_) and the R package “MuMIn” for model selection [68,69].

## 3. Results

### 3.1. Sites and Variables

The literature search produced 67 papers (Appendix A) reporting assemblage and prevalence data falling within our constraints with a total of 162 unique sites in Asia, Europe, North America, and South America (25, 17, 88, and 32, respectively) that were used for analysis. The abundance range (20 ≤ *n* ≤ 200) was determined to capture the most normal distribution of assemblage abundances (mean = 102; median = 55; SD = 176). Significant differences existed between the MPD, MNTD, and PD, (MPD:MNTD t_161_ = −7.31, *p* < 0.001; MPD:PD t_161_ = 36.54, *p* < 0.001; MNTD:PD t_161_ = 29.18, *p* < 0.001). Therefore, all measures of phylogenetic relatedness were used as individual descriptors in the analysis. However, though the mean Chao estimates (ACE and CHAO1) of species richness based from raw abundance (n) were each different from raw richness (s) (t_161_ = 6.86, *p* < 0.001; t_161_ = 4.87, *p* < 0.001, respectively) and from each other (t_161_ = 4.69, *p* < 0.001). These estimates often inflated site richness or were incalculable altogether. Thus, we used the raw species richness value reported in the literature in our dataset. Variables accounting for compressed phylogenetic and diversity characteristics are shown in Figure 1. As expected, the PCA of the assemblage diversity characteristics indicated strong correlations between (s), (PIE), and (H’) (Figure 1a). This can be attributed to the relationship and mathematical influence these metrics may share in their calculation. However, the phylogenetic assemblage descriptors appeared to have independence (Figure 1b).

### 3.2. Model Selection

Model selection (Table 1) indicates the top model accounting for TAS (R^2^_adj_ = 0.12; *p* < 0.001) and includes the dominance index (BP) and ordinate rank (rk) of infected individuals significantly contributing to the prediction of infection in assemblages phylogenetically characterized by the mean nearest taxon distance (MNTD; Table 1). Moreover, the ordinate rank of the most dominant host and the mean nearest taxon distance of species comprising the assemblage were selected as the top predictors of TAS in eight of the eleven (73%) chosen by stepwise linear regression. Further, the MNTD was selected as a top predictor in all 11 models and was shown to be significant in paired with the BP in eight of the eleven (Table 1). Two PCA compressed ecological assemblage descriptors (diversity1; diversity2) and one PCA compressed phylogenetic association descriptor (phylo1) were also important predictors of hantavirus antibody prevalence.

### 3.3. Species Diversity

Overall, 225 small mammal species representing six taxonomic orders and 13 families were reported in our findings (Appendix A). Non-rodent hosts were generally rare, except for shrews and moles (Appendix A), with most of the species being from the order Rodentia, and many of these species being known to serve as hantavirus hosts [5]. Among the rodents, 47% (89/191) of species captured are known hantavirus hosts, with 55% (64/116) of these being cricetid and 63% (19/30) being murid species (Appendix A).

Phylogenetic diversity was negatively correlated with prevalence (Figure 2). Though the correlations between TAS and the phylogenetic predictors MNTD, MPD, and MPD were not strong (R^2^ = 0.03, *p* = 0.02; R^2^ = 0.05, *p* < 0.01; R^2^ = 0.08, *p* < 0.001, respectively), nonetheless the relationships were statistically significant, thus suggesting that assemblages with greater phylogenetic distances between species could be expected to have a lower prevalence. Additionally, species richness (s) was shown to have a negative correlation with TAS (R^2^ = 0.07, *p* < 0.001), though this metric was independently influential in approximately half of the selected models (Table 1; Figure 3).

## 4. Discussion

The dilution effect appears to be a response present in several pathogen systems [70], although the generality of this relationship has been contested [48,71,72,73,74]. In some cases, an increase in the number of species has been associated to a higher prevalence of a pathogen, thus being labeled as an “amplification effect” [71]. Several factors have been proposed as determinants for the occurrence of either effect (i.e., dilution vs amplification), including the ecology of the pathogen, host community composition, and the scale used for examination of the relationship [70,71,75]. Hence, if a unified theory that generates predictions is to be generated across systems, it is necessary to uncover the mechanistic explanations and circumstances in which biodiversity affects pathogen prevalence [40]. However, for most pathogen systems these relationships have not been addressed in detail, particularly in directly transmitted diseases [76].

Much of the research on the mechanics behind the response between diversity and disease prevalence has been done on vector-borne pathogen systems such as Lyme disease. In these vector- mediated systems, differential host competencies and vector preferences create a dilution effect when most members of the community are lost and the remaining hosts are often competent reservoirs contributing to increased transmission events [77]. As a model system for directly transmitted diseases, hantaviruses have been shown to display the dilution effect [78,79], but this response is not universal for all hantavirus systems [48,72,73,74].

Recent evidence of the mechanics present in hantavirus systems has shown that species diversity can act differently on the main drivers of disease transmission (i.e., host density, contact rates, transmissibility). Since these are competing mechanisms, they can cause concurrent increases or decreases in pathogen transmission, with the net effect resulting from the differential strength between opposing mechanisms. A potential explanation why the dilution effect sometimes occurs and sometimes does not is that the quality of the small mammal assemblage may drive hantavirus dynamics [40]. In view of this, a multifactorial relationship between biodiversity and hantavirus transmission is not surprising, as the number of host species at a given site may not be as relevant as species identity in determining prevalence across many sites [48]. The type of component species at a given site represents another aspect of biodiversity that has been little explored within the context of how diversity affects disease dynamics in natural systems. Since the evolutionary legacy of a species sets boundaries to the way it interacts with the environment and other species, we expected this factor to be of relevance for pathogen maintenance at a given host assemblage.

As we hypothesized, our models suggest that the phylogenetic relatedness among small mammal species comprising assemblages plays a role in predicting hantavirus infection prevalence. Greater PD, MPD, and MNTD values correspond to increased assemblage diversity and suggest a general negative trend in the prevalence of hantavirus infection (Figure 2). While this trend is also present with higher species richness (Figure 3), species richness alone does not efficiently explain predictions of antibody prevalence in contrast to models that consider phylogenetic diversity.

A phylo-diverse assemblage is comprised of species less phylogenetically related in an evolutionary context [63], while species richness is a metric used to count only the number of organisms at a locality determined to have their own taxonomic division [53]. Therefore, these metrics represent very different implications for pathogen transmission within a community. Richness in species does not account for the inherent competence potentials shared among species which are closely related [73,80]. Additionally, assemblages of closely related species very likely create assemblage structures (i.e., ranked abundance distributions) dissimilar to assemblages composed of species with varied phylogenetic backgrounds. This suggests that the identity of each species and their ecological contribution to the relative abundance assemblage patterns are more probable to directly impact the strongest drivers (i.e., host densities) of overall infection prevalence.

Beyond the diversity of species as a factor for pathogen prevalence, the species identity and host competency are other relevant aspects of assemblage structure that have been shown to play a critical role in the maintenance of hantaviruses [48,80]. When hosts that are inadequate for the propagation of pathogens (i.e., non-competent hosts) become infected, they are often unable to infect other individuals [81,82]. These non-competent, dead-end hosts tend to be phylogenetically distant from the primary host species [83]. Though a decrease in infection prevalence is often attributed to an increase in species richness, this assumption often disregards the zoonotic potential of other rodent species comprising an assemblage beyond traditionally focused reservoir species [84]. Consequently, a phylogenetic dilution effect may better explain hantavirus transmission dynamics where species phylogenetic relationships more accurately affect the probabilities of localized infections.

Our models support a cautious use of describing dilution effects using only the total species number in an assemblage, while requiring consideration that prevalence trends may be driven by the phylogenetic relatedness of species comprising the assemblage. Though localized assemblages may have a relatively high richness of species present (e.g., *Peromyscus* rodents) this is not necessarily indicative of a diverse assemblage when groups of species are phylogenetically similar [48,63,73,85]. Each of our top models contains predictor variables associated with phylogenetic diversity, and a majority (9/11) include the ordinate rank of the most dominant host in the assemblage (Table 1). These data support our hypotheses that, overall, more phylogenetically diverse assemblages tend to have lower hantavirus antibody prevalence and a higher species richness can influence a decrease in prevalence in some cases (Figure 2 and Figure 3).

Though, in general, the R^2^ values are not high, a pertinent consideration is the heterogeneity of the data used to conduct the analyses. While steps were taken to standardize the dataset extracted from the literature, there are obvious differences embedded in the methods, techniques, sampling effort and other factors that created a heterogeneous dataset. However, despite the inherent variation created by the intrinsic nature of the dataset, what is worth emphasizing is the consistent response of phylogenetic variables as important among the contrasted models. Though exploratory, our study indicates that a potential avenue to move forward in the diversity versus pathogen prevalence debate is to ascertain the effects of the phylogenetic legacy of species on the main drivers (i.e., host density, contact rates, and transmissibility) of disease transmission [40].

Though it may be unusual to find a high prevalence in diverse assemblages, it is important to consider the species interactions and spillover maintenance of hantavirus infection within the context of community scale. Predictions of hantavirus infection prevalence may not be entirely reliant upon the dominance of specific known host species (e.g., rodents), as these viruses can be maintained within small mammal communities with high diversity. For example, two soricomorph species, (*Suncus murinus* and *Urotrichus talpoides*) were the only hantavirus antibody positive species in rodent-dominated assemblages in Vietnam [86] and Japan [12]. In the case of *S. murinus*, this was the first report for the Seewis Virus [11], and Arai et al. [12] reported the Asama virus as the first recognized mole-born hantavirus which was carried by *U. talpoides.* The highest hantavirus diversity was found in *S. murinus*. This shrew species was found to carry Hantaan [87], Seoul [88], and Thottapalayam viruses [86] suggesting that viral host-switching can be a fundamental driver of hantavirus maintenance [89].

Anthropogenic changes to habitats can create patches which often favor population increases of phylogenetically related rodent species, many of which are hantavirus reservoir species [90,91]. These environmental alterations can increase the risk of human interaction with infected rodents [9,92,93,94]. Patch networks have been shown to artificially increase the number of different potential hosts, thereby creating an artificial localized increase in species richness [95,96,97] while decreasing species diversity [39,40,72,79,80,98].

Knowledge of small mammal assemblage structure is fundamental in predicting the prevalence of hantavirus infection. The models we presented describe the general relationship between host assemblage characteristics and the influence of species’ phylogenetic relatedness to predict prevalence. These factors, concerted with species’ abundance and dominance characteristics, can provide valuable insight into the hantavirus system and may contribute to the prevention of human hantaviral infection in changing landscapes.

## Figures and Tables

**Figure 1 viruses-11-00671-f001:**
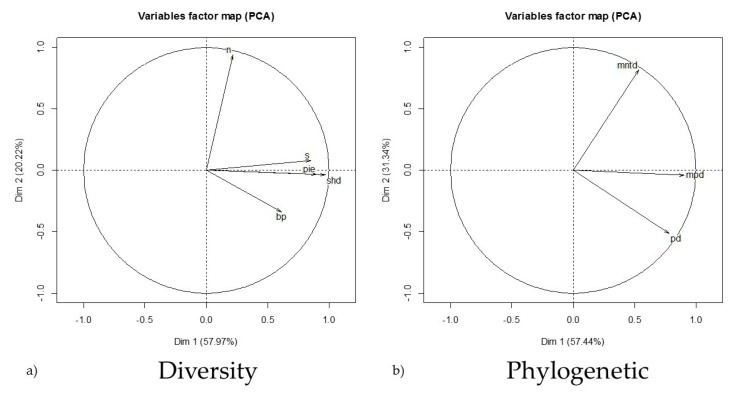
The results of covariate collapsing using principal component analysis (PCA) to derive eigenvectors describing small mammal assemblage diversity characteristics where dimensions one and two account for ~58% and ~20% of the variation, respectively (**a**), and phylogenetic relatedness indices where dimensions one and two account for ~57% and ~31% of the variation, respectively (**b**). These descriptors were used as predictor variables in the global model selection to account for variation in predicting the total assemblage orthohantavirus prevalence at the global scale. (n) abundance; (s) species richness; (pie) evenness; (shd) diversity index; (bp) dominance index; (mntd) mean nearest taxon distance; (mpd) mean pairwise distance; (pd) phylogenetic distance.

**Figure 2 viruses-11-00671-f002:**
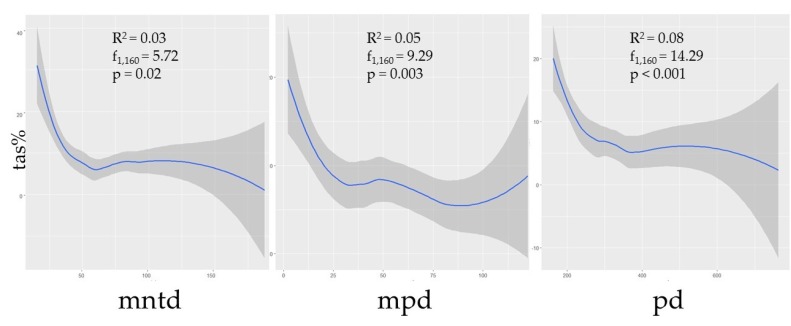
The curveplots show the relationship of total assemblage seroprevalence (TAS) against three phylogenetic indices: Mean nearest taxon distance (MNTD), mean pairwise distance (MPD), and phylogenetic diversity (PD). Confidence intervals (95%) are shown in grey. Data are shown at the global scale.

**Figure 3 viruses-11-00671-f003:**
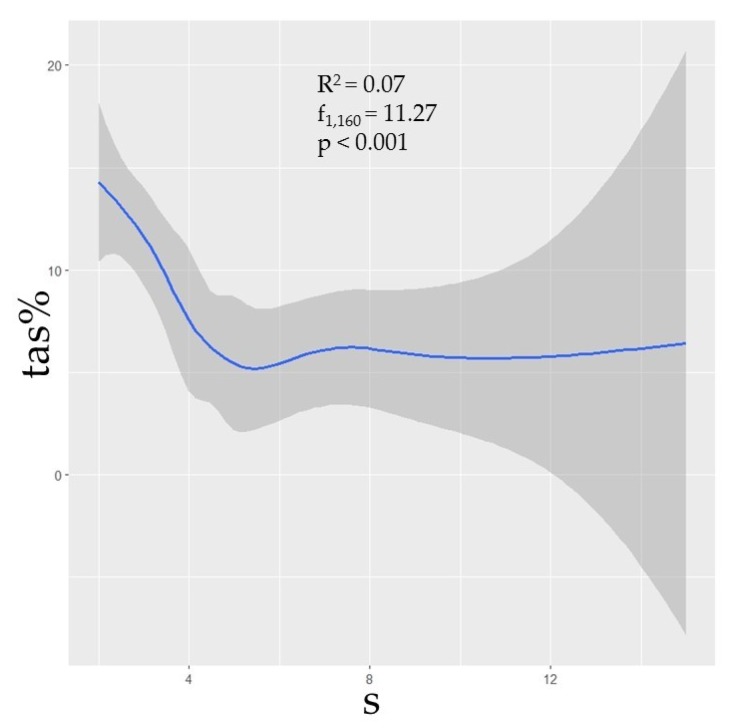
The curveplots show the relationship of total assemblage seroprevalence (TAS) against raw species richness (s). Confidence intervals (95%) shown in grey. Data are shown at the global scale.

**Table 1 viruses-11-00671-t001:** Model selection using Akaike Information Criterion (AIC_c_) to predict hantavirus prevalence in small mammal assemblages at the global scale. The models suggest the identity of each species (s) and phylogenetic relatedness (mntd; mpd; pd) of dominant hosts (bp; rank) are informative predictors of assemblage prevalence (tas). Principal component analysis was used to avoid multicolinearity of covariates (diversity1; diversity2; phylo1). Models are listed in order of decreasing comparative weights.

Model	df	logLik	AIC_c_	Δ	Weight	R^2^	p
tas~ (s ***) + (bp *) + (rank *) + (mntd **)	6	−577.401	1167.3	0.00	0.372	0.12	<0.001
tas~ (s **) + pie + (bp *) + (rank *) + (mntd **)	7	−577.152	1169.0	1.69	0.160	0.12	<0.001
tas~ (diversity2 ***) + (bp *) + rank + (mntd **)	6	−578.418	1169.4	2.03	0.135	0.11	<0.001
tas~ diversity2 + (bp *) + rank + (mntd *) + pd	7	−577.534	1168.8	2.45	0.109	0.11	<0.001
tas~ diversity2 + (bp *) + rank + (mntd *) + mpd + pd	8	−576.943	1170.8	3.48	0.065	0.11	<0.001
tas~ n + (s **) + pie + (bp *) + rank + (mntd **)	8	−576.955	1170.9	3.51	0.064	0.11	<0.001
tas~ n + (phylo1 **) + mntd	5	−580.731	1171.8	4.50	0.039	0.09	<0.001
tas~ n+(s **) + pie + (bp *) + rank + mpd + (mntd *)	9	−576.796	1172.8	5.43	0.025	0.11	<0.001
tas~ n + diversity1 + phylo1 + mntd	6	−580.493	1173.5	6.18	0.017	0.08	0.001
tas~ n + s + pie + (bp *) + rank + mpd + pd + (mntd *)	10	−576.778	1175.0	7.67	0.008	0.1	0.002
tas~ n + diversity1 + rank + phylo1 + mntd	7	−580.452	1175.6	8.29	0.006	0.08	0.003

Within-model parameter significance codes: (***) p~0.0; (**) p~0.01; (*) p~0.05.

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
