# Peer review of "The Ecology and Phylogeny of Hosts Drive the Enzootic Infection Cycles of Hantaviruses"

_viruses, 2019, doi:10.3390/v11070671_

Round 1

Reviewer 1 Report

The paper describes a very useful model in predicting hantavirus diseases in rodents based on environmental and biological information. This is a very interesting idea, as hantaviruses are widespread viruses and rodents serve as a "popular" hantavirus hosts and source of infection. 

The introduction of the paper is very informative, shows the importance of the study due to the frequency of hantaviruses infections. 

I do not feel an expert to evaluate the methods used, but from the text of this manuscript I believe I understand the idea, what makes me believe that this will be comprehensive to the readers. The results being shown on figures and in table are clear.

According to me, the best part of the paper is the discussion, where Authors discuss the evidences for the usage of the models.

The Authors concluded that knowledge of a small mammal assemblage structure is fundamental in predicting prevalence of hantavirus infection. I think that this conclusion may be also a first step to concluding on other viruses with similar etiology.

I recommend the paper to be published.

Author Response

We would like to thank you for spending time with our manuscript, and offering such positive feedback. Please see attached PDF with our changes listed in the track changes.

Reviewer 2 Report

Milholland et al., developed predictive models for prevalence of hantavirus infection in small mammal  assemblages to test the hypothesis that small mammal assemblages with more phylogenetically diverse species would have fewer hantavirus antibody prevalent individuals.  The authors obtained data points for their models from the literature, ran these through the model and proved their hypotheses. Overall, the paper will well written but needs to be cleaned up a bit. The significance of the work is in the calibration of the models with data from the literature.  

Major issues:

1.       More explanation of figure 1 is required, especially for readers not familiar with this type of analysis.

Minor issues:

1.       Line 48. There appears to be a missing word just in front of the citations.

2.       Line 53. Would Intraspecies transmission be better terminology and “within-species transmission”

3.       Line 60-61. The sentence “However, ecological …. predictive models.” Should be rewritten for clarity.

4.       Line 186, the sites add up to 162 not 164.

5.       Line 220. Missing number for p value

6.       Line 318 Although is misspelled  

7.       There seems to be a figure missing from the next to the last page of the pdf. There are a and b listed on the page but nothing else.

Author Response

We would like to thank you for spending time with our manuscript, and offering such positive feedback. Your suggestions were very helpful and provided a valuable perspective. We have made changes per your advice. Please see attached PDF with our changes listed in the track changes.

Reviewer 3 Report

The authors present a thorough investigation into the available literature regarding hantavirus prevalence in small mammal assemblages in an attempt to ultimately predict prevalence of hantavirus infection in humans.  While the methods and associated results are valid, the manuscript is geared more toward a biostatistical audience.  It may be a difficult read for the majority of individuals who read articles from this journal.  Nonetheless, the authors present an interesting approach to understanding a problem that is likely to increase in severity as humans expand into the ecology of small mammals harboring hantavirus.

There are minor grammatical errors throughout.  For example:

Line 48 - the sentence abruptly ends

Line 58 - "infections" is stated twice

Line 73 - "ocurr" is spelled wrong

Line 78 - "phylogentic" is spelled wrong

Line 188 - "difference" is singular and should be plural

Line 244 - "effect" is stated twice

There are others not mentioned here; it is suggested that the authors perform a thorough review for grammatical errors.

Lines 185-186 specify 164 unique sites, but the numbers specified (25, 17, 88, and 32) add up to 162.  Please clarify and update data as needed.

Table 1 is very confusing.  I do not have suggestions on how to improve it, but it is too busy and difficult to follow.

Author Response

(The authors gave the same response as above.)
